# An Improved Calculation Formula of the Extended Entropic Chaos Degree and Its Application to Two-Dimensional Chaotic Maps

**DOI:** 10.3390/e23111511

**Published:** 2021-11-14

**Authors:** Kei Inoue

**Affiliations:** Faculty of Engineering, Sanyo-Onoda City University, 1-1-1 Daigaku-Dori, Sanyo-Onoda 756-0884, Yamaguchi, Japan; kinoue@rs.socu.ac.jp

**Keywords:** chaos, Lyapunov exponent, extended entropic chaos degree

## Abstract

The Lyapunov exponent is primarily used to quantify the chaos of a dynamical system. However, it is difficult to compute the Lyapunov exponent of dynamical systems from a time series. The entropic chaos degree is a criterion for quantifying chaos in dynamical systems through information dynamics, which is directly computable for any time series. However, it requires higher values than the Lyapunov exponent for any chaotic map. Therefore, the improved entropic chaos degree for a one-dimensional chaotic map under typical chaotic conditions was introduced to reduce the difference between the Lyapunov exponent and the entropic chaos degree. Moreover, the improved entropic chaos degree was extended for a multidimensional chaotic map. Recently, the author has shown that the extended entropic chaos degree takes the same value as the total sum of the Lyapunov exponents under typical chaotic conditions. However, the author has assumed a value of infinity for some numbers, especially the number of mapping points. Nevertheless, in actual numerical computations, these numbers are treated as finite. This study proposes an improved calculation formula of the extended entropic chaos degree to obtain appropriate numerical computation results for two-dimensional chaotic maps.

## 1. Introduction

The Lyapunov exponent (LE) is a widely used measure for quantifying the chaos of a dynamical system. However, it is generally incomputable for time series. Therefore, some estimation methods for the Lyapunov exponent of a time series have been suggested in previous studies [1,2,3,4,5,6]. However, it is well-known that estimating the Lyapunov exponent for a time series is difficult.

The entropic chaos degree (ECD) was introduced to measure the chaos of a dynamical system in the field of information dynamics [7]. The ECD is directly computable, even for time series data obtained from dynamical systems. Some researchers have sought to characterize certain chaotic behaviors using the ECD [8,9,10]. Recently, it was demonstrated that the modified ECD coincides with the Lyapunov exponent for a one-dimensional chaotic map under typical chaotic conditions [11,12]. Moreover, the extended entropic chaos degree (EECD) was shown to be the sum of all the Lyapunov exponents of a multidimensional chaotic map under typical chaotic conditions [13]. However, it was assumed that the number of mapping points and the number of all components of equipartition of the domain are infinity. In actual computations, these numbers are treated as finite numbers. In this study, I aim to formulate a calculation such that the EECD is also equal to the sum of all the Lyapunov exponents of two-dimensional typical chaotic maps in actual numerical computations.

In this study, I propose an improved calculation formula of the EECD for multidimensional chaotic maps. Moreover, I apply the improved calculation formula of the EECD to two-dimensional typical chaotic maps.

## 2. Entropic Chaos Degree

In this section, I briefly review the definition of the ECD for a difference equation system,
$$x_{n+1}=f\left(x_{n}\right),\;n=0,1,\ldots ,$$
where *f* represents a map such that f:I→I ($\equiv {\left[a,b\right]}^{d}\subset R^{d},\;a,b\in R,\;d\in N$).

Let $x_{0}$ represent an initial value and $\left\{A_{i}\right\}$ represent a finite partition of *I* such that
$$I=\overset{N}{\underset{k=1}{⋃}}A_{k},\;\;A_{i}\cap A_{j}=\emptyset \;\left(i\neq j\right).$$
Next, the probability distribution $\left(p_{i,A}^{\left(n\right)}\left(M\right)\right)$ at time *n* and joint distribution $\left(p_{i,j,A}^{\left(n,n+1\right)}\left(M\right)\right)$ at time *n* and n+1 associated with the difference equation are expressed as follows:$$\begin{array}{ccc} p_{i,A}^{\left(n\right)}\left(M\right) & = & \frac{1}{M}\sum_{k=n}^{n+M-1}1_{A_{i}}\left(x_{k}\right)\\ & = & \frac{\left|\left\{x_{k}\in A_{i};\;n\leq k\leq n+M-1\right\}\right|}{M},\\ p_{i,j,A}^{\left(n,n+1\right)}\left(M\right) & = & \frac{1}{M}\sum_{k=n}^{n+M-1}1_{A_{i}}\left(x_{k}\right)1_{A_{j}}\left(x_{k+1}\right)\\ & = & \frac{\left|\left\{\left(x_{k},x_{k+1}\right)\in A_{i}\times A_{j};\;n\leq k\leq n+M-1\right\}\right|}{M}, \end{array}$$
where $1_{A}$ represents the characteristic function of a set *A*.

The ECD *D* of the orbit $\left\{x_{n}\right\}$ is then defined in [7] as follows:(1)$$\begin{array}{ccc} D^{\left(M,n\right)}\left(A,f\right) & = & \sum_{i=1}^{N}\sum_{j=1}^{N}p_{i,j,A}^{\left(n\right)}\left(M\right)\log\frac{p_{i,A}^{\left(n\right)}\left(M\right)}{p_{i,j,A}^{\left(n,n+1\right)}\left(M\right)}\\ & = & \sum_{i=1}^{N}p_{i,A}^{\left(n\right)}\left(M\right)\left(-\sum_{j=1}^{N}p_{A}^{\left(n\right)}\left(j|i\right)\left(M\right)\log p_{A}^{\left(n\right)}\left(j|i\right)\left(M\right)\right), \end{array}$$
where
$$p_{A}^{\left(n\right)}\left(j|i\right)\equiv \frac{p_{i,j,A}^{\left(n,n+1\right)}\left(M\right)}{p_{i,A}^{\left(n\right)}\left(M\right)}$$
represents the conditional probability from the component $A_{i}$ of $\left\{A_{i}\right\}$ to the component $A_{j}$ of $\left\{A_{i}\right\}$.

Further, the ECD is denoted as $D^{\left(M\right)}\left(A,f\right)$ without *n* if the orbit $\left\{x_{n}\right\}$ does not depend on time *n*. Moreover, the ECD is denoted as $D^{\left(M,n\right)}\left(A\right)$ without *f* if the map *f* does not produce the orbit $\left\{x_{n}\right\}$.

The ECD is larger than the Lyapunov exponent for a one-dimensional chaotic map [12].

At the end of this section, I discuss the relation between the ECD and the metric entropy. For sufficiently large *M*, there exists a probability measure μ on *I* without depending on *n*. Let (X,A,μ) be a measure space with μ(X)=1. For provided measurable partitions ξ and ζ of *X*, the conditional entropy $H_{\mu }\left(\xi \left|\zeta \right.\right)$ of ξ with respect to ζ is defined in [14] by
(2)$$H_{\mu }\left(\xi \left|\zeta \right.\right)=-\sum_{C\in \xi ,D\in \zeta }\mu \left(C\cap D\right)\log\frac{\mu \left(C\cap D\right)}{\mu \left(D\right)}.$$
If T:I→I is a measurable transformation preserving a probability measure μ on *I* then, for sufficiently large *M*, I have
$$\begin{array}{ccc} D^{\left(M\right)}\left(\xi ,T\right) & ≃ & -\sum_{C\in \xi }\mu \left(C\cap T(C)\right)\log\frac{\mu \left(C\cap T(C)\right)}{\mu \left(T(C)\right)}\\ & = & -\sum_{C\in \xi }\mu \left(C\cap T(C)\right)\log\frac{\mu \left(C\cap T(C)\right)}{\mu \left(C\right)}\\ & = & -\sum_{C\in \xi }\mu \left(T^{-1}\left(C\cap T(C)\right)\right)\log\frac{\mu \left(T^{-1}\left(C\cap T(C)\right)\right)}{\mu \left(T^{-1}\left(C\right)\right)}\\ & = & -\sum_{C\in \xi }\mu \left(T^{-1}\left(C\right)\cap C\right)\log\frac{\mu \left(T^{-1}\left(C\right)\cap C\right)}{\mu \left(T^{-1}\left(C\right)\right)}\\ & = & H_{\mu }\left(\xi \left|T^{-1}\xi \right.\right)\\ & \geq & \lim_{n\to \infty }H_{\mu }\left(\xi \left|\overset{n}{\underset{i=1}{⋁}}T^{-i}\xi \right.\right). \end{array}$$
In the last inequality, I used the property such that if ζ is a refinement of η, then $H_{\mu }\left(\xi |\zeta \right)\leq H_{\mu }\left(\xi |\eta \right)$ for every partition ξ, where $\eta =T^{-1}\xi$ and $\zeta ={⋁}_{i=1}^{n}T^{-i}\xi$.

Then the metric entropy $h_{\mu }\left(T,\xi \right)$ of *T* with respect to μ and a measurable partition ξ has the following property [14].
$$h_{\mu }\left(T,\xi \right)=H_{\mu }\left(\xi \left|\overset{n}{\underset{i=1}{⋁}}T^{-i}\xi \right.\right).$$
Therefore, I obtain
$$D^{\left(M\right)}\left(\xi ,T\right)\geq h_{\mu }\left(T,\xi \right)$$
for sufficiently large *M*.

## 3. Extended Entropic Chaos Degree

In this section, it is assumed that
(3)$$N=L^{d},\;\;I=\prod_{l=1}^{d}\left[a_{l},b_{l}\right].$$
Let the $L^{d}$-equipartitions $\left\{A_{i}\right\}$ of *I* be
$$I=\overset{L^{d}-1}{\underset{k=0}{⋃}}A_{k}.$$

For any component $A_{i}$ of $\left\{A_{i}\right\}$, I divide another component $A_{j}$ into ${\left(S_{i,j}\right)}^{d}$-equipartitions ${\left\{B_{l}^{\left(i,j\right)}\right\}}_{0\leq l\leq {\left(S_{i,j}\right)}^{d}-1}$ of smaller components, such that
(4)$$A_{j}=\overset{{\left(S_{i,j}\right)}^{d}-1}{\underset{l=0}{⋃}}B_{l}^{\left(i,j\right)}.$$
For each $B_{l}^{\left(i,j\right)}$, the function $g_{i,j}$ is defined as follows:(5)$$g_{i,j}\left(B_{l}^{\left(i,j\right)}\right)=\left\{\begin{array}{cc} 1 & \left(B_{l}^{\left(i,j\right)}\cap f\left(A_{i}\right)\neq \emptyset \right)\\ 0 & \left(B_{l}^{\left(i,j\right)}\cap f\left(A_{i}\right)=\emptyset \right) \end{array}\right.$$
Using the function $g_{i,j}$, for any two components $A_{i},A_{j}\;\left(i\neq j\right)$ of the initial partition $\left\{A_{i}\right\}$, the function $R(S_{i,j})$ is defined as follows:$$R\left(S_{i,j}\right)=\frac{\sum_{l=0}^{{\left(S_{i,j}\right)}^{d}-1}g_{i,j}\left(B_{l}^{\left(i,j\right)}\right)}{{\left(S_{i,j}\right)}^{d}}.$$
The EECD $D_{S}$ is provided in [13] as follows:$$\begin{array}{ccc} D_{S}^{\left(M,n\right)}\left(A,f\right) & = & \sum_{i=0}^{L^{d}-1}p_{i,A}^{\left(n\right)}\left(M\right)\sum_{j=0}^{L^{d}-1}p_{A}^{\left(n\right)}\left(j|i\right)\left(M\right)\log\frac{R(S_{i,j})}{p_{A}^{\left(n\right)}\left(j|i\right)\left(M\right)}, \end{array}$$
where $S={\left(S_{i,j}\right)}_{0\leq i,j\leq L^{d}-1}$.

Note that the EECD $D_{S}$ becomes the CD, as shown in Equation (Equation 1), only if $R(S_{i,j})=1$ for any two components $A_{i}$ and $A_{j}$ of the initial partition $\left\{A_{i}\right\}$.

First, the following theorem concerning the periodic orbit is presented [13].

**Theorem** **1.**
*Let*

L,M

*represent sufficiently large natural numbers. If map f creates a stable periodic orbit with period T, the following equality holds.*

(6)
$$D_{S}^{\left(M,n\right)}\left(A,f\right)=-\frac{d}{T}\sum_{k=1}^{T}\log S_{i_{k},j_{k}}.$$



Second, I briefly review the relationship between the EECD and the Lyapunov exponent in a chaotic dynamical system. Let a map *f* be a piecewise $C^{1}$ function on $R^{d}$. For any $x={\left(x_{1},x_{2},\ldots ,x_{d}\right)}^{t}$, $y={\left(y_{1},y_{2},\ldots ,y_{d}\right)}^{t}$ $\in A_{i}$, I consider an approximate Jacobian matrix $\hat{J}$ as follows:$$\hat{J}\left(x,y\right)={\left(\frac{f_{i}\left(x\right)-f_{i}\left(y\right)}{x_{j}-y_{j}}\right)}_{1\leq i,j\leq d}.$$
Let $r_{k}\left(x,y\right)$(k=1,2,…,d) represent the eigenvalues of $\sqrt{\hat{J^{t}}\left(x,y\right)\hat{J}\left(x,y\right)}$.

Then, the following properties are assumed to be satisfied.

**Assumption** **1.**
*For sufficiently large natural numbers, L and M, I assume that the following conditions are satisfied.*

*(1) Points in*

$A_{i}$

*are uniformly distributed over*

$A_{i}$

*.*

*(2) Then,*

$r_{k}\left(x,y\right)=r_{k}^{\left(i\right)},\;k=1,2,\ldots ,d$

*is obtained for any*

$x,y\in A_{i}$

*.*



Next, the following theorem is presented [13].

**Theorem** **2.**
*For any*

$A_{i},\;i=0,1,\ldots ,L^{d}-1$

*, Assumption 1 is assumed to be satisfied. Then, the following is obtained.*

$$\lim_{S\to \infty }\lim_{L\to \infty }\lim_{M\to \infty }D_{S}^{\left(M,m\right)}\left(A,f\right)=\sum_{k=1}^{d}\lambda_{k},$$

*where*

$$S\to \infty \Leftrightarrow S_{i,j}\to \infty \;\left(i,j=0,1,\ldots ,L^{d}-1\right)$$

*and*

$\left\{\lambda_{1},\ldots ,\lambda_{d}\right\}$

*represent the Lyapunov spectrum of a map f.*


Theorem 2 implies that if the points on any $A_{i}$ are uniformly distributed, then the EECD becomes the sum of all the Lyapunov exponents of the map *f* in the limits at infinity of *M*, *L*, and $S_{i,j}$.

At the end of this section, I discuss the relationship between the EECD and the metric entropy. For sufficiently large *M*, a probability measure μ exists on *I* without depending on *n*. If T:I→I is a measurable transformation preserving a probability measure μ on *I*, then for sufficiently large *M* and $S_{i,j}$, I have
$$\begin{array}{ccc} D_{S}^{\left(M,n\right)}\left(\xi ,T\right) & ≃ & \sum_{C\in \xi }\mu \left(C\cap T(C)\right)\log\frac{\frac{m\left(C\cap T(C)\right)}{m\left(C\right)}}{\frac{\mu \left(C\cap T(C)\right)}{\mu \left(T(C)\right)}}\\ & = & \sum_{C\in \xi }\mu \left(C\cap T(C)\right)\log\frac{\frac{m\left(C\cap T(C)\right)}{m\left(C\right)}}{\frac{m\left(C\cap T(C)\right)}{m\left(C\right)}}\\ & = & \sum_{C\in \xi }\mu \left(C\cap T(C)\right)\log1\\ & = & 0. \end{array}$$
Here, *m* is the Lebesgue measure on $R^{d}$.

Because $h_{\mu }\left(T,\xi \right)\geq 0$, I have
$$D_{S}^{\left(M\right)}\left(\xi ,T\right)\leq h_{\mu }\left(T,\xi \right)$$
for sufficiently large $M,S_{i,j}$.

## 4. Improvement of Calculation Formula of the Extended Entropic Chaos Degree

In Theorem 2, it is assumed that the values of *L*, *M*, and $S_{i,j}$ are equal to infinity. However, in actual numerical computations, these numbers are treated as finite numbers. I propose an improved calculation formula of the EECD to obtain appropriate numerical computation results.

First, I consider improving a calculation formula of the EECD when the map *f* creates a stable periodic orbit. If the map *f* creates a stable periodic orbit, then, for any component $A_{i}$ with $A_{i}\neq \emptyset$, there exists a component $A_{j_{i}}$ such that
$$|A_{j_{i}}\cap f\left(A_{i}\right)|=|f\left(A_{i}\right)\left|=\right|A_{j_{i}}|.$$
It follows that
(7)$$p_{A}^{\left(n\right)}\left(j|i\right)=\left\{\begin{array}{cc} 1 & \left(j=j_{i}\right)\\ 0 & \left(j\neq j_{i}\right) \end{array}\right..$$
From Equation (Equation 7), I obtain
$$D_{S}^{\left(M,n\right)}\left(A,f\right)=\sum_{|A_{i}|>0}p_{i,A}^{\left(n\right)}\left(M\right)\log R\left(S_{i,j_{i}}\right).$$
Now, for any component $A_{i}$, let us consider $A_{j}$ such that $A_{j}\cap f\left(A_{i}\right)\neq \emptyset$. Let $C_{i,j}$ be the number of $B_{l}^{\left(i,j\right)}$ such that $B_{l}^{\left(i,j\right)}\cap f\left(A_{i}\right)\neq \emptyset$, that is,
$$C_{i,j}\equiv \left|\left\{B_{l}^{\left(i,j\right)}:\;\left(x_{k},f\left(x_{k}\right)\right)\in A_{i}\times B_{l}^{\left(i,j\right)},\;l=0,1,\ldots ,{\left(S_{i,j}\right)}^{d}-1\right\}\right|,$$
where
$$A_{j}=\overset{{\left(S_{i,j}\right)}^{d}-1}{\underset{l=0}{⋃}}B_{l}^{\left(i,j\right)}.$$
When the map *f* creates a stable periodic orbit, I set
$$\left(S_{i,j_{i}}\right)=\left\lfloor \sqrt[{d}]{\left|A_{i}\right|}\right\rfloor .$$
I then have
$$R\left(S_{i,j_{i}}\right)=\frac{C_{i,j}}{{\left(S_{i,j_{i}}\right)}^{d}}≃\frac{1}{|A_{i}|}≃\frac{\left|\left\{A_{i}:\;\left|A_{i}\right|>0\right\}\right|}{M}.$$
Thus, when the map *f* creates a stable periodic orbit, I use
(8)$$\begin{array}{ccc} {\tilde{D}}_{S,1}^{\left(M,n\right)}\left(A,f\right) & \equiv & \sum_{|A_{i}|>0}p_{i,A}^{\left(n\right)}\left(M\right)\log\frac{\left|\left\{A_{i}:\;\left|A_{i}\right|>0\right\}\right|}{M}\\ & = & \log\frac{\left|\left\{A_{i}:\;\left|A_{i}\right|>0\right\}\right|}{M} \end{array}$$
to calculate the EECD.

Second, I consider improving a calculation formula of the EECD when the map *f* does not create a periodic orbit. For any sufficiently large natural numbers, *L* and *M*, let us assume the conditions (1) and (2) in Assumption 1. Let *m* be the Lebesgue measure on $R^{d}$ and μ be the invariant measure of *f*. Then, I obtain
(9)$$\begin{array}{ccc} D_{S}^{\left(M,n\right)}\left(A,f\right) & = & \sum_{i=0}^{L^{d}-1}p_{i,A}^{\left(n\right)}\left(M\right)\left(\sum_{j=0}^{L^{d}-1}p_{A}^{\left(n\right)}\left(j|i\right)\left(M\right)\log\frac{R(S_{i,j})}{p_{A}^{\left(n\right)}\left(j|i\right)\left(M\right)}\right)\\ & ≃ & \sum_{i=0}^{L^{d}-1}\mu \left(f\left(A_{i}\right)\right)\left(\sum_{j=0}^{L^{d}-1}\frac{\mu (A_{j}\cap f\left(A_{i}\right))}{\mu (f\left(A_{i}\right))}\log\frac{\frac{m(A_{j}\cap f\left(A_{i}\right))}{m(A_{j})}}{\frac{\mu (A_{j}\cap f\left(A_{i}\right))}{\mu (f\left(A_{i}\right))}}\right)\\ & ≃ & \sum_{i=0}^{L^{d}-1}\sum_{j=0}^{L^{d}-1}\mu \left(A_{j}\cap f\left(A_{i}\right)\right)\log\frac{m(f\left(A_{i}\right))}{m(A_{j})}. \end{array}$$
Here, the second approximation (Equation (Equation 9)) uses the following:$$\frac{\mu \left(A_{j}\cap f\left(A_{i}\right)\right)}{\mu \left(f(A_{i})\right)}≃\frac{m\left(A_{j}\cap f\left(A_{i}\right)\right)}{m\left(f(A_{i})\right)}.$$
Then, I directly obtain the following: (10)$$\begin{array}{c} \sum_{i=0}^{L^{d}-1}\sum_{j=0}^{L^{d}-1}\mu \left(A_{j}\cap f\left(A_{i}\right)\right)\log\frac{m(f\left(A_{i}\right))}{m(A_{j})}\\ =\sum_{i=0}^{L^{d}-1}\mu \left(f\left(A_{i}\right)\right)\log m\left(f\left(A_{i}\right)\right)-\sum_{j=0}^{L^{d}-1}\mu \left(A_{j}\right)\log m\left(A_{j}\right)\\ ≃\sum_{i=0}^{L^{d}-1}p_{i,A}^{\left(n\right)}\left(M\right)\log m\left(f\left(A_{i}\right)\right)-\sum_{i=0}^{L^{d}-1}p_{i,A}^{\left(n\right)}\left(M\right)\log m\left(A_{i}\right)\\ =\sum_{i=0}^{L^{d}-1}p_{i,A}^{\left(n\right)}\left(M\right)\log\frac{m(f\left(A_{i}\right))}{m(A_{i})}. \end{array}$$
Now, for any set $X\;\left(\neq \emptyset \right)\subset I=\prod_{k=1}^{d}\left[a_{k},b_{k}\right]$,
$$\begin{array}{ccc} X & = & \left\{\left(x_{1},x_{2},\ldots ,x_{d}\right):\;x_{k}\in \left[a_{k},b_{k}\right],k=1,2,\ldots ,d\right\}\\ & = & \left\{\left({\left(x_{1}\right)}_{j},{\left(x_{2}\right)}_{j},\ldots ,{\left(x_{d}\right)}_{j}\right):\;{\left(x_{k}\right)}_{j}\in \left[a_{k},b_{k}\right],k=1,2,\ldots ,d,j=0,1,\ldots ,\left|X\right|-1\right\}. \end{array}$$
The variance–covariance matrix $\sum_{X}$ to all points x on *X* is given by
$$\sum_{X}=\left(\begin{array}{cccc} {\left(\sigma_{1}^{2}\right)}_{X} & {\left(\sigma_{1,2}\right)}_{X} & \ldots & {\left(\sigma_{1,d}\right)}_{X}\\ {\left(\sigma_{2,1}\right)}_{X} & {\left(\sigma_{2}^{2}\right)}_{X} & \ldots & {\left(\sigma_{2,d}\right)}_{X}\\ \vdots & \vdots & \ddots & \vdots \\ {\left(\sigma_{d,1}\right)}_{X} & {\left(\sigma_{d,2}\right)}_{X} & \ldots & {\left(\sigma_{d}^{2}\right)}_{X} \end{array}\right),$$
where
$$\begin{array}{ccc} {\left(\sigma_{l,m}\right)}_{X} & = & \frac{1}{\left|X\right|}\sum_{j=0}^{|X|-1}\left({\left(x_{l}\right)}_{j}-{\bar{x}}_{l}\right)\left({\left(x_{m}\right)}_{j}-{\bar{x}}_{m}\right),\\ {\left(\sigma_{l}^{2}\right)}_{X} & = & \frac{1}{\left|X\right|}\sum_{j=0}^{|X|-1}{\left({\left(x_{l}\right)}_{j}-{\bar{x}}_{l}\right)}^{2},\\ {\left({\bar{x}}_{l}\right)}_{X} & = & \frac{1}{\left|X\right|}\sum_{j=0}^{|X|-1}{\left(x_{l}\right)}_{j}. \end{array}$$
Let ${\left(\lambda_{k}\right)}_{X}\;\left(k=1,2,\ldots ,d\right)$ be eigenvalues of $\sum_{X}$ such that ${\left(\lambda_{i}\right)}_{X}\geq {\left(\lambda_{j}\right)}_{X}\;\left(i\geq j\right)$. For any sufficiently large natural numbers, *L* and *M*, I have
(11)$$\frac{m(f\left(A_{i}\right))}{m(A_{i})}≃\frac{2^{d}\prod_{k=1}^{d}\sqrt{{\left(\lambda_{k}\right)}_{f(A_{i})}}}{2^{d}\prod_{k=1}^{d}\sqrt{{\left(\lambda_{k}\right)}_{A_{i}}}}=\frac{\prod_{k=1}^{d}\sqrt{{\left(\lambda_{k}\right)}_{f(A_{i})}}}{\prod_{k=1}^{d}\sqrt{{\left(\lambda_{k}\right)}_{A_{i}}}}.$$
From Equations (Equation 10) and (Equation 11), when the map *f* does not create a periodic orbit, I use
(12)$$\begin{array}{ccc} {\tilde{D}}_{S,2}^{\left(M,n\right)}\left(A,f\right) & \equiv & \sum_{|A_{i}|>0}p_{i,A}^{\left(n\right)}\left(M\right)\log\frac{\prod_{k=1}^{d}\sqrt{{\left(\lambda_{k}\right)}_{f(A_{i})}}}{\prod_{k=1}^{d}\sqrt{{\left(\lambda_{k}\right)}_{A_{i}}}} \end{array}$$
as the calculation formula of the EECD.

Let ${\left(u_{k}\right)}_{X}$ be the eigenvector corresponding to the eigenvalue ${\left(\lambda_{k}\right)}_{X}$, and
$${\langle x\rangle }_{X}\equiv {\left({\bar{x}}_{1},{\bar{x}}_{2},\ldots ,{\bar{x}}_{d}\right)}_{X}.$$
In actual numerical computations, let us consider subsets $C_{i},D_{i}$ of $A_{i},f\left(A_{i}\right)$ such that
(13)$$\begin{array}{ccc} C_{i} & = & \left\{{\langle x\rangle }_{A_{i}}+\sum_{k=1}^{d}\alpha_{k}\sqrt{{\left(\lambda_{k}\right)}_{A_{i}}}\frac{{\left(u_{k}\right)}_{A_{i}}}{∥{\left(u_{k}\right)}_{A_{i}}∥}:\;-1\leq \alpha_{k}\leq 1\right\}, \end{array}$$
(14)$$\begin{array}{ccc} D_{i} & = & \left\{{\langle x\rangle }_{f(A_{i})}+\sum_{k=1}^{d}\beta_{k}\sqrt{{\left(\lambda_{k}\right)}_{f(A_{i})}}\frac{{\left(u_{k}\right)}_{f(A_{i})}}{∥{\left(u_{k}\right)}_{f(A_{i})}∥}:\;-1\leq \beta_{k}\leq 1\right\}. \end{array}$$
Now, I assume that all points x on $A_{i},f\left(A_{i}\right)$ are almost uniformly distributed over $C_{i},D_{i}$, such that
$$\frac{|E_{i}|}{|C_{i}|}≃\frac{m(E_{i})}{m(C_{i})},\;\frac{|F_{i}|}{|D_{i}|}≃\frac{m(F_{i})}{m(D_{i})}$$
for any subsets $E_{i},F_{i}$ of $C_{i},D_{i}$. Then, I obtain
(15)$$\frac{m(f\left(C_{i}\right))}{m(C_{i})}=\frac{m(D_{i})}{m(C_{i})}≃\frac{2^{d}\prod_{k=1}^{d}\sqrt{{\left(\lambda_{k}\right)}_{f(A_{i})}}}{2^{d}\prod_{k=1}^{d}\sqrt{{\left(\lambda_{k}\right)}_{A_{i}}}}=\frac{\prod_{k=1}^{d}\sqrt{{\left(\lambda_{k}\right)}_{f(A_{i})}}}{\prod_{k=1}^{d}\sqrt{{\left(\lambda_{k}\right)}_{A_{i}}}}.$$
Moreover, I denote the eigenvalues of $\sqrt{Df^{t}\left(x\right)Df\left(x\right)}$ such that $r_{i}\left(x\right)\geq r_{j}\left(x\right)$
(i≥j) by $r_{k}\left(x\right)$
(k=1,2,…,d). Then, I have
(16)$$\begin{array}{ccc} {\tilde{D}}_{S,2}^{\left(M,n\right)}\left(A,f\right) & ≃ & \sum_{|C_{i}|>0}p_{i,A}^{\left(n\right)}\left(M\right)\log\frac{m(f\left(C_{i}\right))}{m(C_{i})}\\ & = & \sum_{|C_{i}|>0}\int_{C_{i}}\log\left(\prod_{k=1}^{d}r_{k}\left(x\right)\right)p\left(x\right)\prod_{l=1}^{d}dx_{l}\\ & = & \int_{a_{1}}^{b_{1}}\int_{a_{2}}^{b_{2}}\cdots \int_{a_{d}}^{b_{d}}\log\left(\prod_{k=1}^{d}r_{k}\left(x\right)\right)p\left(x\right)\prod_{l=1}^{d}dx_{l}\\ & = & \sum_{k=1}^{d}\int_{a_{1}}^{b_{1}}\int_{a_{2}}^{b_{2}}\cdots \int_{a_{d}}^{b_{d}}\log\left(r_{k}\left(x\right)\right)p\left(x\right)\prod_{l=1}^{d}dx_{l}\\ & = & \sum_{k=1}^{d}\lambda_{k}. \end{array}$$
Here, p(x) is the density function of x and $\left\{\lambda_{1},\lambda_{2},\ldots ,\lambda_{d}\right\}$ is the Lyapunov spectrum of *f*.

In the sequel, I use
(17)$${\tilde{D}}_{S}^{\left(M,n\right)}\left(A,f\right)\equiv \left\{\begin{array}{cc} {\tilde{D}}_{S,1}^{\left(M,n\right)}\left(A,f\right) & \left(when\;the\;map\;f\;creates\;a\;stable\;periodic\;orbit\right)\\ {\tilde{D}}_{S,2}^{\left(M,n\right)}\left(A,f\right) & \left(otherwise\right) \end{array}\right.$$
as the calculation formulas of the EECD.

## 5. Numerical Computation Results of the EECD for Two-Dimensional Chaotic Maps

In this section, I apply the improved calculation formulas (Equation (Equation 12)) of the EECD to two-dimensional typical chaotic maps. In the sequel, I set *M* = 1,000,000 and $L=\sqrt{M}=$ 1000. (In principle, the double type in C language is used in numerical computations. However, the floating-point type with its 1024-bit mantissa is used in numerical calculations of eigenvalues of the variance–covariance matrix by GMP (GNU Multi-Precision Library).)

Let us consider the generalized baker’s map $f_{a}$ as a simple two-dimensional dissipative chaotic map such that the Jacobian matrix $Df_{a}\left(x\right)$ does not depend on x.

The generalized baker’s map $f_{a}$ is defined by
(18)$$f_{a}\left(x\right)=\left\{\begin{array}{cc} \left(2ax_{1},\frac{1}{2}ax_{2}\right)\; & \left(0\leq x_{1}\leq \frac{1}{2}\right)\\ \left(a\left(2x_{1}-1\right),\frac{1}{2}a\left(x_{2}+1\right)\right)\; & \left(\frac{1}{2}<x_{1}\leq 1\right), \end{array}\right.$$
where $x={\left(x_{1},x_{2}\right)}^{t}\in \left[0,1\right]\times \left[0,1\right]$ and 0≤a≤1.

The generalized baker’s map $f_{a}$ for 0.5≤a≤1.0 corresponds to the following operations: first, the unit square is stretched 2a times in the $x_{1}$ direction and compressed a/2 times in the $x_{2}$ direction; second, the right part protruding from the unit square is cut vertically and stacked on the top of the left part. The first operation is called “stretching” and the second operation is called “folding”. These two operations are essential basic elements for producing chaotic behaviors.

### 5.1. Numerical Computation Results of the EECD for Generalized Baker’s Map

The Jacobian matrix of the baker’s map $f_{a}$ is expressed as follows:(19)$$Df_{a}\left(x\right)=\left(\begin{array}{cc} 2a & 0\\ 0 & \frac{1}{2}a \end{array}\right).$$
Thus, $Df_{a}\left(x\right)$ depends only on the parameter *a*. The dynamics produced by the baker’s map $f_{a}$ is dissipative for 0≤a<1 because $|detDf_{a}\left(x\right)|=a^{2}$.

For $e_{1}={\left(1,0\right)}^{t},\;e_{2}={\left(0,1\right)}^{t}$, I obtain
(20)$${\hat{e}}_{1}\equiv Df_{a}\left(e_{1}\right)e_{1}=2ae_{1},\;{\hat{e}}_{2}\equiv Df_{a}\left(e_{2}\right)e_{2}=\frac{1}{2}ae_{2}.$$
Thus, the expansion rate in the stretching of the baker’s map $f_{a}$ is 2a and the contraction rate in the folding of the baker’s map $f_{a}$ is a/2. I then consider the orbit $\left\{x_{n}\right\}$ produced by the generalized baker’s map $f_{a}$, as follows:$$x_{n+1}=f_{a}\left(x_{n}\right),\;n=0,1,2,\ldots ,x_{0}={\left(0.3333,0.3333\right)}^{t}.$$

First, I present typical orbits of the baker’s map $f_{a}$ in Figure 1. As the parameter *a* increases, the spread of points is mapped from a linear distribution to the entire unit square.

Second, I present the numerical computation results of the LEs $\lambda_{1},\lambda_{2}$$\left(\lambda_{1}>\lambda_{2}\right)$, the total sum $\lambda_{1}+\lambda_{2}$ of the LEs, the ECD *D*, and the EECD ${\tilde{D}}_{S}$ of the baker’s map $f_{a}$ in Figure 2. Figure 2 shows that the EECD ${\tilde{D}}_{S}$ takes approximately the exact value of the total sum $\lambda_{1}+\lambda_{2}$ of the LEs for the generalized baker’s map $f_{a}$.

In general, the orthogonal basis of $R^{d}$ can be changed by *f*. In the sequel, for a two-dimensional chaotic map *f*, I consider the average expansion rate in the stretching of *f* as $\exp(\lambda_{1})$ and the average contraction rate in the folding of *f* as $\exp(\lambda_{2})$, where $\lambda_{1},\lambda_{2}$ are the LEs of *f* such that $\lambda_{1}>0>\lambda_{2}$.

### 5.2. Numerical Computation Results of the EECD for Tinkerbell Map

Let us consider the Tinkerbell map $f_{a}$ as a two-dimensional dissipative chaotic map such that the Jacobian matrices $Df_{a}\left(x\right)$ and $\det Df_{a}\left(x\right)$ depend on x and the parameter *a*.

The Tinkerbell map $f_{a}$ is defined by
(21)$$f_{a}\left(x\right)={\left(x_{1}^{2}-x_{2}^{2}+ax_{1}-0.6013x_{2},2x_{1}x_{2}+2x_{1}+0.5x_{2}\right)}^{t},$$
where $x={\left(x_{1},x_{2}\right)}^{t}\in \left[a_{1},b_{1}\right]\times \left[a_{2},b_{2}\right]$.

For 0.7≤a≤0.9, I obtain the following:$$a_{1}=-1.3,\;\;a_{2}=-1.6,\;\;b_{1}=0.5,\;\;b_{2}=0.6.$$

The Jacobian matrix of the Tinkerbell map $f_{a}$ is expressed as follows:(22)$$Df_{a}\left(x\right)=\left(\begin{array}{cc} 2x_{1}+a & -2x_{2}-0.6013\\ 2x_{2}+2 & 2x_{1}+0.5 \end{array}\right).$$
Thus, $Df_{a}\left(x\right)$ depends on x and the parameter *a*.

I then consider the orbit $\left\{x_{n}\right\}$ produced by the Tinkerbell map $f_{a}$ as follows:$$x_{n+1}=f_{a}\left(x_{n}\right),\;n=0,1,2,\ldots ,x_{0}={\left(0.1,0.1\right)}^{t}.$$

First, I present typical orbits of the Tinkerbell map $f_{a}$ in Figure 3. The orbit of the Tinkerbell map $f_{a}$ constructs a strange attractor at a=0.9. The map $f_{a}$ is named the Tinkerbell map because the shape of the attractor produced by the Tinkerbell map looks like the movement of a fairy named Tinker Bell, who appears in a Disney film.

Second, I present the numerical computation results of the LEs $\lambda_{1},\lambda_{2}$ $\left(\lambda_{1}>\lambda_{2}\right)$, the total sum $\lambda_{1}+\lambda_{2}$ of the LEs, the ECD *D*, and the EECD ${\tilde{D}}_{S}$ of the Tinkerbell map $f_{a}$ in Figure 4. Figure 4 shows that the EECD ${\tilde{D}}_{S}$ takes almost the same value as the total sum $\lambda_{1}+\lambda_{2}$ of the LEs for the Tinkerbell map $f_{a}$ at most *a* for 0.7≤a≤0.9. However, the Tinkerbell map $f_{a}$ creates a stable periodic orbit at several *a*’s. Then the EECD takes a different value from the total sum $\lambda_{1}+\lambda_{2}$ of LEs for the Tinkerbell map $f_{a}$ because I use ${\tilde{D}}_{S,1}^{\left(M,n\right)}$(Equation (Equation 8)) as the calculation formula of the EECD ${\tilde{D}}_{S}$.

### 5.3. Numerical Computation Results of the EECD for Ikeda Map

Let us consider the Ikeda map $f_{a}$ as a two-dimensional dissipative chaotic map such that the Jacobian matrix $Df_{a}\left(x\right)$ depends on x and the parameter *a* but that $\det Df_{a}\left(x\right)$ does not depend on x.

The modified Ikeda map is given as the complex map in [15,16]
(23)$$f\left(z\right)=A+Bze^{iK/(|z{|}^{2}+1)+C},\;z\in C,\;\;A,B,K,C\in R.$$
The Ikeda map $f_{a}$ is defined as a real two-dimensional example of Equation (Equation 23) by
(24)$$f_{a}\left(x\right)={\left(1+a\left(x_{1}\cos t-x_{2}\sin t\right),a\left(x_{1}\sin t+x_{2}\cos t\right)\right)}^{t},$$
where
$$t=0.4-\frac{6}{1+x_{1}^{2}+x_{2}^{2}}$$
and $x={\left(x_{1},x_{2}\right)}^{t}\in \left[a_{1},b_{1}\right]\times \left[a_{2},b_{2}\right]$.

For 0.7≤a≤0.9, I obtain the following:$$a_{1}=-0.4,\;\;a_{2}=-2.3,\;\;b_{1}=1.8\;\;,b_{2}=0.9.$$

The Jacobian matrix of the Ikeda map $f_{a}$ is expressed as follows:(25)$$Df_{a}\left(x\right)=a\left(\begin{array}{cc} u_{1}\cos t-u_{2}\sin t & -u_{3}\sin t-u_{4}\cos t\\ u_{1}\sin t+u_{2}\cos t & u_{3}\cos t-u_{4}\sin t \end{array}\right),$$
where
$$\begin{array}{ccc} & & u_{1}=1-\frac{12x_{1}x_{2}}{{\left(1+x_{1}^{2}+x_{2}^{2}\right)}^{2}},\;\;u_{2}=\frac{12x_{1}^{2}}{{\left(1+x_{1}^{2}+x_{2}^{2}\right)}^{2}}\\ & & u_{3}=1+\frac{12x_{1}x_{2}}{{\left(1+x_{1}^{2}+x_{2}^{2}\right)}^{2}},\;\;u_{4}=\frac{12x_{2}^{2}}{{\left(1+x_{1}^{2}+x_{2}^{2}\right)}^{2}}. \end{array}$$
Thus, $Df_{a}\left(x\right)$ depends on x and the parameter *a*. The dynamics produced by the Ikeda map $f_{a}$ are dissipative for 0≤a<1 because $|\det Df_{a}\left(x\right)|=a^{2}$.

I then consider the orbit $\left\{x_{n}\right\}$ produced by the Ikeda map $f_{a}$ as follows:$$x_{n+1}=f_{a}\left(x_{n}\right),\;n=0,1,2,\ldots ,x_{0}={\left(0.1,0.0\right)}^{t}.$$

First, I present typical orbits of the Ikeda map $f_{a}$ in Figure 5. As the parameter *a* increases, the attractor constructed by the Ikeda map $f_{a}$ becomes larger. Regarding $f_{a}$ plots, the Ikeda map might be conjugated to a Hénon map [17].

Second, let us assume that $dv_{0}$ is transformed to $dv_{m}$ by $f_{a}^{m}$ on $R^{2}$. For the Ikeda map $f_{a}$, using the chain rule and $\det Df_{a}\left(x\right)=a^{2}$ at any x, I have
(26)$$dv_{m}=\det Df_{a}^{m}\left(v_{0}\right)dv_{0}=a^{2m}dv_{0}.$$
Therefore, I obtain
(27)$$\lambda_{1}+\lambda_{2}=\lim_{m\to \infty }\frac{1}{m}\log\left|\frac{dv_{m}}{dv_{0}}\right|=\lim_{m\to \infty }\frac{\log a^{2m}}{m}=2\log a,$$
where $\lambda_{k}$(k=1,2) are the LEs of the Ikeda map $f_{a}$ such that $\lambda_{1}>\lambda_{2}$.

I present the numerical computation results of the LEs $\lambda_{1},\lambda_{2}$, the total sum $\lambda_{1}+\lambda_{2}$ of the LEs, the ECD *D*, and the EECD ${\tilde{D}}_{S}$ of the Ikeda map $f_{a}$ in Figure 6. Figure 6 shows that the EECD ${\tilde{D}}_{S}$ takes almost the same value as the total sum $\lambda_{1}+\lambda_{2}$ of the LEs for the Ikeda map $f_{a}$ at almost *a* for 0.7≤a≤0.9. However, the Ikeda map $f_{a}$ creates a stable periodic orbit at several *a*’s. Then the EECD takes a different value from the total sum $\lambda_{1}+\lambda_{2}$ of LEs for the Ikeda map $f_{a}$ because I use ${\tilde{D}}_{S,1}^{\left(M,n\right)}$(Equation (Equation 8)) as the calculation formula of the EECD ${\tilde{D}}_{S}$.

### 5.4. Numerical Computation Results of the EECD for Hénon Map

Let us consider the Hénon map $f_{a,b}$ as a two-dimensional dissipative chaotic map such that the Jacobian matrix $Df_{a,b}\left(x\right)$ depends on x and the parameter *b* but that the Jacobian $\det Df_{a,b}\left(x\right)$ does not depend on x.

The Hénon map $f_{a,b}$ is expressed as follows:(28)$$f_{a,b}\left(x\right)={\left(a-x_{1}^{2}+bx_{2},x_{1}\right)}^{t},$$
where $x={\left(x_{1},x_{2}\right)}^{t}\in \left[a_{1},b_{1}\right]\times \left[a_{2},b_{2}\right]$.

For a=1.4,0<b≤0.3, I obtain the following:$$a_{k}=-1.8,\;\;b_{k}=1.8,\;\left(k=1,2\right).$$
In the sequel, we rewrite $f_{1.4,b}=f_{b}$.

The Jacobian matrix of the Hénon map $f_{a,b}$ is expressed as follows:(29)$$Df_{a,b}\left(x\right)=\left(\begin{array}{cc} 2x_{1} & b\\ 1 & 0 \end{array}\right).$$
Thus, $Df_{a,b}\left(x\right)$ depends on $x_{1}$ and the parameter *b*. The dynamics produced by the Hénon map $f_{a,b}$ are dissipative for 0≤b<1 because $|\det Df_{a,b}\left(x\right)|=b$.

I then consider the orbit $\left\{x_{n}\right\}$ produced by the Hénon map $f_{b}$ as follows:$$x_{n+1}=f_{b}\left(x_{n}\right),\;n=0,1,2,\ldots ,x_{0}={\left(0.1,0.1\right)}^{t}.$$

First, I present typical orbits of the Hénon map $f_{b}$ in Figure 7. The orbit of the Hénon attractor has a fractal structure. Expanding a strip region, I find that innumerable parallel curves reappear in the strip.

Second, let us assume that $dv_{0}$ is transformed to $dv_{m}$ by $f_{b}^{m}$ on $R^{2}$. For the Hénon map $f_{b}$, using the chain rule and $\det Df_{b}\left(x\right)=-b$ at any x, I have
(30)$$dv_{m}=\det Df_{b}^{m}\left(v_{0}\right)dv_{0}={\left(-b\right)}^{m}dv_{0}.$$
Therefore, I obtain
(31)$$\lambda_{1}+\lambda_{2}=\lim_{m\to \infty }\frac{1}{m}\log\left|\frac{dv_{m}}{dv_{0}}\right|=\lim_{m\to \infty }\frac{\log b^{m}}{m}=\log b,$$
where $\lambda_{k}$(k=1,2) are the LEs of the Hénon map $f_{b}$ such that $\lambda_{1}>\lambda_{2}$.

I present the numerical computation results of the LEs $\lambda_{1},\lambda_{2}$$\left(\lambda_{1}>\lambda_{2}\right)$, the total sum $\lambda_{1}+\lambda_{2}$ of the LEs, the ECD *D*, and the EECD ${\tilde{D}}_{S}$ for the Hénon map $f_{b}$ in Figure 8. Figure 8 shows that the EECD ${\tilde{D}}_{S}$ takes a value almost equal to the total sum $\lambda_{1}+\lambda_{2}$ of the LEs for the Hénon map $f_{b}$ at most *b* for 0.1<b≤0.3. However, the EECD takes a different value from the total sum $\lambda_{1}+\lambda_{2}$ of LEs for the Hénon map $f_{a}$, even though the Hénon map $f_{a}$ does not create a periodic orbit at many *b*s for 0<b≤0.1. Here, the absolute value of the negative LE $\lambda_{2}$ is much larger than the absolute value of the positive LE $\lambda_{1}$.

Now, let $\rho_{A_{i}}$ be the autocorrelation function to all points x on a component $A_{i}$. I consider the average of $|\rho_{A_{i}|}$ such that
(32)$$E\left(\left|\rho \right|\right)=\sum_{|A_{i}|>3}\frac{|A_{i}|}{\sum_{|A_{i}|>3}\left|A_{i}\right|}\left|\rho_{A_{i}}\right|.$$
I present the numerical computation results of the total sum $\lambda_{1}+\lambda_{2}$ of the LEs, the EECD ${\tilde{D}}_{S}$, and the average of $|\rho_{A_{i}}|$ for the Hénon map $f_{b}$ in Figure 9.

Here, at d=2, the denominator of the right side of Equation (Equation 11) is given by
(33)$$\sqrt{{\left(\lambda_{1}\right)}_{A_{i}}{\left(\lambda_{2}\right)}_{A_{i}}},$$
where ${\left(\lambda_{k}\right)}_{A_{i}}$(k=1,2) is the eigenvalue of the variance–covariance matrix $\sum_{A_{i}}$ to all points x on $A_{i}$.

Let ${\left(\sigma_{k}^{2}\right)}_{A_{i}}\;\left(k=1,2\right)$ and ${\left(\sigma_{1,2}\right)}_{A_{i}}$ be the variances and covariance of all points on $A_{i}$, respectively. Then, I have
$$\begin{array}{ccc} {\left(\lambda_{1}\right)}_{A_{i}} & = & \frac{{\left(\sigma_{1}^{2}\right)}_{A_{i}}+{\left(\sigma_{2}^{2}\right)}_{A_{i}}+\sqrt{{\left\{{\left(\sigma_{1}^{2}\right)}_{A_{i}}+{\left(\sigma_{2}^{2}\right)}_{A_{i}}\right\}}^{2}-4{\left(\sigma_{1}^{2}\right)}_{A_{i}}{\left(\sigma_{2}^{2}\right)}_{A_{i}}\left\{1-{\left(\rho_{A_{i}}\right)}^{2}\right\}}}{2},\\ {\left(\lambda_{2}\right)}_{A_{i}} & = & \frac{{\left(\sigma_{1}^{2}\right)}_{A_{i}}+{\left(\sigma_{2}^{2}\right)}_{A_{i}}-\sqrt{{\left\{{\left(\sigma_{1}^{2}\right)}_{A_{i}}+{\left(\sigma_{2}^{2}\right)}_{A_{i}}\right\}}^{2}-4{\left(\sigma_{1}^{2}\right)}_{A_{i}}{\left(\sigma_{2}^{2}\right)}_{A_{i}}\left\{1-{\left(\rho_{A_{i}}\right)}^{2}\right\}}}{2}. \end{array}$$
Therefore, if the absolute value of $\rho_{A_{i}}$ is equal to 1, then I have
(34)$${\left(\lambda_{1}\right)}_{A_{i}}={\left(\sigma_{1}^{2}\right)}_{A_{i}}+{\left(\sigma_{2}^{2}\right)}_{A_{i}},\;{\left(\lambda_{2}\right)}_{A_{i}}=0.$$
Thus, it becomes difficult to estimate $m\left(f\left(A_{i}\right)\right)/m\left(A_{i}\right)$ by Equation (Equation 11) when the absolute value of $\rho_{A_{i}}$ is approximately 1. Therefore, the EECD takes a different value from the total sum of the LEs when E(|ρ|) is near 1.

### 5.5. Numerical Computation Results of the EECD for Standard Map

Let us consider the standard map $f_{K}$ as a two-dimensional conservative chaotic map such that the Jacobian matrix $Df_{K}\left(y\right)$ depends on y and the parameter *K*.

The standard map $f_{K}$ is defined as follows:(35)$$f_{K}\left(y\right)={\left(\theta +p+K\sin\theta ,\;p+K\sin\theta \right)}^{t},$$
where $y={\left(\theta ,p\right)}^{t}\in {\left[-\pi ,\pi \right]}^{2}$.

The Jacobian matrix of the standard map $f_{K}$ is expressed as follows:(36)$$Df_{K}\left(y\right)=\left(\begin{array}{cc} 1+K\cos\theta & 1\\ K\cos\theta & 1 \end{array}\right).$$
Thus, $Df_{K}\left(x\right)$ depends on θ and the parameter *K*. The dynamics produced by the standard map $f_{K}$ become conservative because $|\det Df_{K}\left(x\right)|=1$.

I then consider the orbit $\left\{y_{n}\right\}$ produced by the standard map $f_{K}$ as follows:$$y_{n+1}=f_{K}\left(y_{n}\right),\;n=0,1,2,\ldots ,y_{0}={\left(1.5,2.0\right)}^{t}.$$

First, I present typical orbits of the standard map $f_{K}$ with initial point $\left(\theta_{0},p_{0}\right)=\left(1.5,2.0\right)$ in Figure 10.

The standard map consists of the Poincaré’s surface of the section of the kicked rotator. The map has a linear structure around K=0. However, as *K* increases, the map produces a nonlinear structure and chaos for an appropriate initial condition.

Second, let us assume that $dv_{0}$ is transformed to $dv_{m}$ by $f_{K}^{m}$ on $R^{2}$. For the standard map $f_{K}$, using the chain rule and $\det Df_{K}\left(x\right)=1$, I have
(37)$$dv_{m}=\det Df_{K}^{m}\left(v_{0}\right)dv_{0}=dv_{0}.$$
Therefore, I obtain
(38)$$\lambda_{1}+\lambda_{2}=\lim_{m\to \infty }\frac{1}{m}\log\left|\frac{dv_{m}}{dv_{o}}\right|=0,$$
where $\lambda_{k}$(k=1,2) are the LEs of the standard map $f_{K}$ such that $\lambda_{1}>\lambda_{2}$.

I present the numerical computation results of the LEs $\lambda_{1},\lambda_{2}$$\left(\lambda_{1}>\lambda_{2}\right)$, the total sum $\lambda_{1}+\lambda_{2}$ of the LEs, the ECD *D*, and the EECD ${\tilde{D}}_{S}$ for the standard map $f_{K}$ in Figure 11. Figure 11 shows that as *K* increases, the difference between the EECD ${\tilde{D}}_{S}$ and the total sum $\lambda_{1}+\lambda_{2}$ of LEs for the standard map $f_{b}$ increases. In other words, as the positive LE increases, the difference between the EECD and the total sum of the LE increases.

Now, I consider symmetric difference equations such that
(39)$$x_{n+1}+x_{n-1}=2x_{n}+K\sin x_{n}.$$
Here, Equation (Equation 39) can arise as a discretization of $\frac{d^{2}}{dt^{2}}x=g\left(x\right)-2x$ with g(x)=2x+Ksinx [18].

Introducing new variables $\theta_{n}\equiv x_{n}$, $p_{n}\equiv x_{n}-x_{n-1}$, Equation (Equation 39) can be written as
(40)$$\begin{array}{c} \theta_{n+1}=\theta_{n}+p_{n}+K\sin\theta_{n},\\ p_{n+1}=p_{n}+K\sin\theta_{n}. \end{array}$$
This mapping is equivalent to the standard map Equation (Equation 35).

Moreover, let *R* be an involution such that $R\left(x_{n},x_{n-1}\right)=\left(x_{n-1},x_{n}\right)$. Then, I have
(41)$$R\left(\theta_{n},p_{n}\right)=\left(\theta_{n}-p_{n},-p_{n}\right).$$
Using ${\left(R\circ f\right)}^{2}=id$ and $R^{2}=id$, I obtain
(42)$$R\circ f=f^{-1}\circ R,$$
which signifies that the standard map $f_{K}$ is reversible with respect to the involution *R*.

Equation (Equation 40) is area preserving as well as reversible, as is common with area-preserving maps [19]. Since the standard map $f_{k}$ is reversible, two Lyapunov exponents of $f_{K}$ become $\lambda_{1}$ and $\lambda_{2}$ such that $\lambda_{1}=-\lambda_{2}>0$ by Theorem 3.2 in [20].

Let us consider increasing and decreasing *L* of the EECD. I represent the numerical computation results of the EECD at L=500,1000,2000 for the standard map $f_{a}$ in Figure 12. Figure 12 shows that as *L* increases, the EECD ${\tilde{D}}_{S}$ goes to the total sum $\lambda_{1}+\lambda_{2}$ of the LEs for the standard map $f_{K}$.

## 6. Conclusions

In this study, I have focused on improving the calculation formula of the EECD and applied the improved calculation formula of the EECD to two-dimensional typical chaotic maps. I have shown that the EECD is almost equal to the total sum of the LEs for their chaotic maps in many cases. However, for the two cases, the EECD was different from the total sum of the LEs even though the map did not create a periodic orbit.

The first case occurs when the absolute value of the negative LE is much larger than the absolute value of the positive LE. Evidently, for the Hénon map $f_{a}$, the EECD takes a much larger value than the total sum of the LE at many *a*’s for 0<a≤0.1. Then, the average E(|ρ|) of the absolute value of the autocorrelation function $\rho_{A_{i}}$ to all points on component $A_{i}$ was approximately one. Here, it becomes difficult to estimate $m\left(f\left(A_{i}\right)\right)/m\left(A_{i}\right)$ by Equation (Equation 11). Therefore, the EECD takes a different value from the total sum of the LEs when E(|ρ|) is approximately one.

The second case occurs notably when the positive LE takes a large value. Evidently, for the standard map $f_{K}$, as the parameter *K* increases, the difference between the EECD and the total sum of the LE increases. In other words, as the positive LE increases, the difference between the EECD and the total sum of the LEs also increases. Here, I have shown the possibility of reducing the above difference by increasing *L*, where $L^{2}$ is the number of equipartitions $\left\{A_{i}\right\}$ of $I={\left[-\pi ,\pi \right]}^{2}$.

I have applied the improved calculation formulas of the EECD to two-dimensional chaotic maps. However, in future works, I will discuss applying the improved calculation formulas of the EECD to higher-dimensional chaotic dynamics.

## Figures and Tables

**Figure 1 entropy-23-01511-f001:**
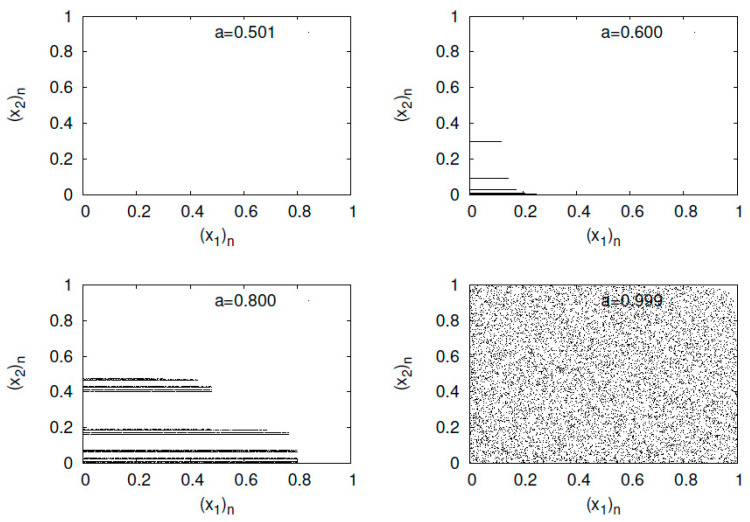
${\left(x_{2}\right)}_{n}$ versus ${\left(x_{1}\right)}_{n}$ for the generalized baker’s map $f_{a}$.

**Figure 2 entropy-23-01511-f002:**
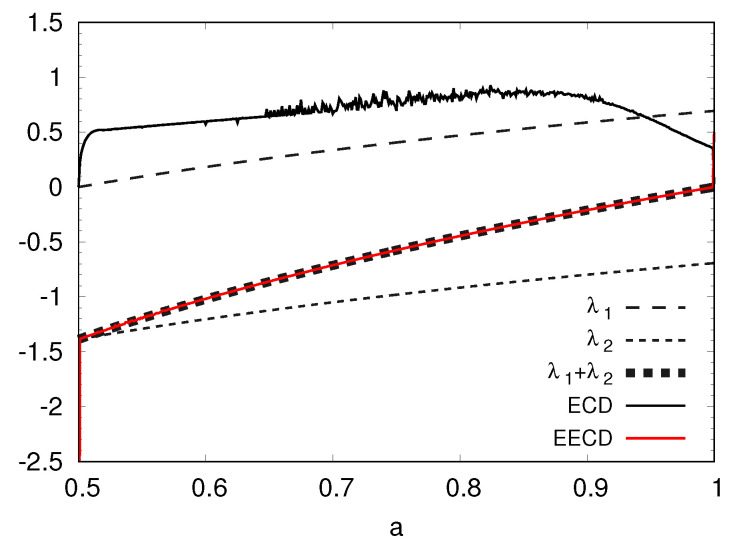
$\lambda_{1}$, $\lambda_{2}$, $\lambda_{1}+\lambda_{2}$, *D*, ${\tilde{D}}_{S}$ versus *a* for the generalized baker’s map $f_{a}$.

**Figure 3 entropy-23-01511-f003:**
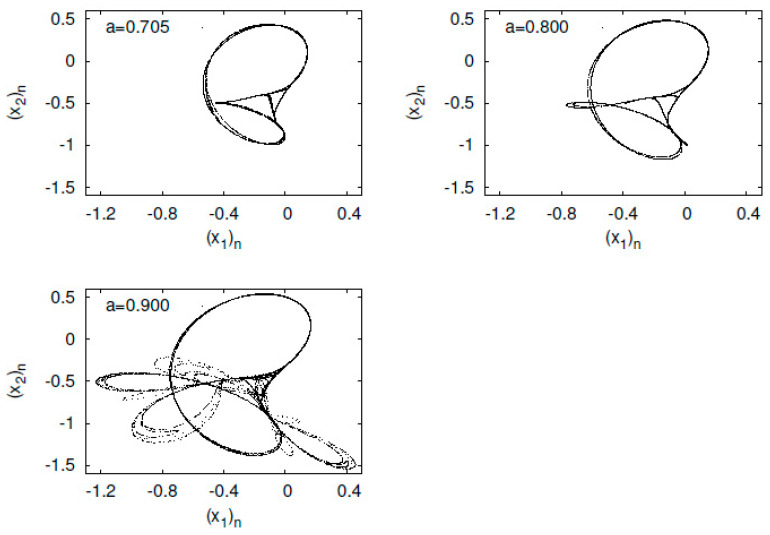
${\left(x_{2}\right)}_{n}$ versus ${\left(x_{1}\right)}_{n}$ for the Tinkerbell map $f_{a}$.

**Figure 4 entropy-23-01511-f004:**
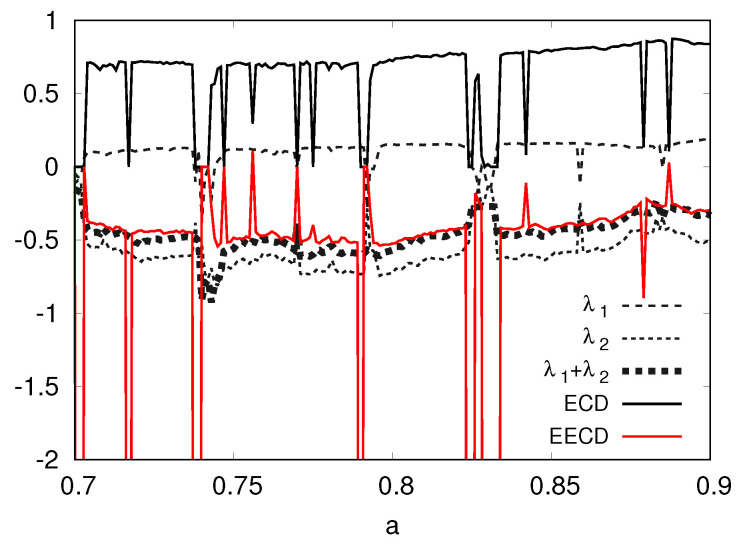
$\lambda_{1},\lambda_{2},\lambda_{1}+\lambda_{2}$, *D*, ${\tilde{D}}_{S}$ versus *a* for the Tinkerbell map $f_{a}$.

**Figure 5 entropy-23-01511-f005:**
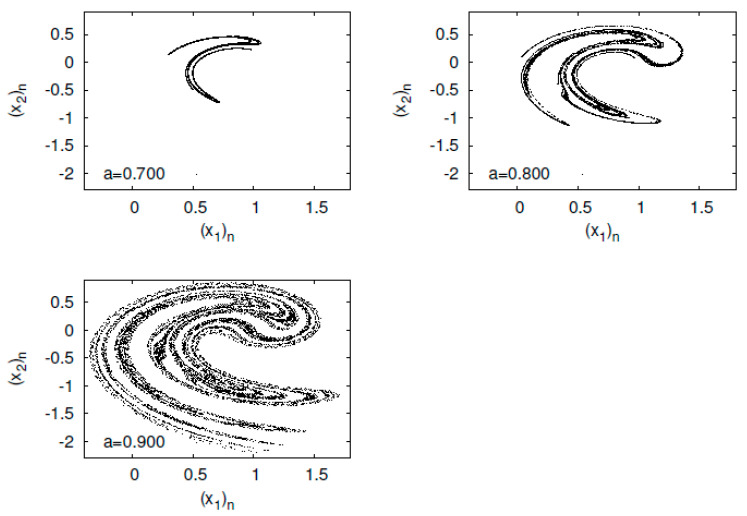
${\left(x_{2}\right)}_{n}$ versus ${\left(x_{1}\right)}_{n}$ for the Ikeda map $f_{a}$.

**Figure 6 entropy-23-01511-f006:**
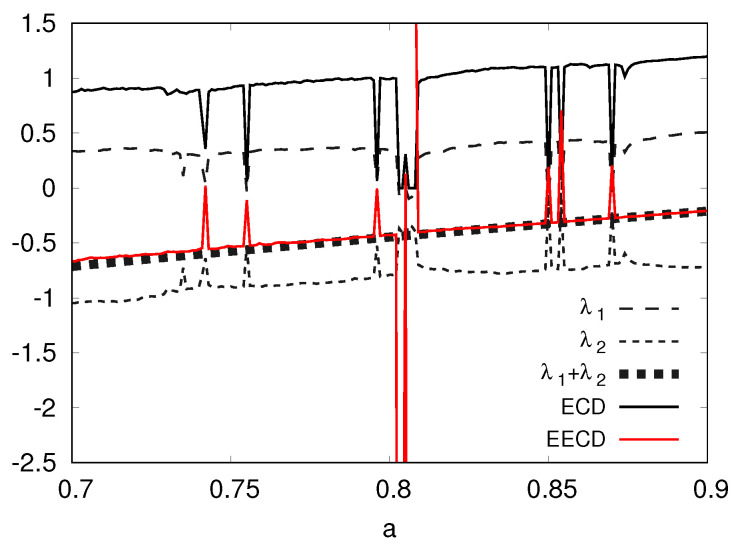
$\lambda_{1},\lambda_{2},\lambda_{1}+\lambda_{2}$, *D*, ${\tilde{D}}_{S}$ versus *a* for the Ikeda map $f_{a}$.

**Figure 7 entropy-23-01511-f007:**
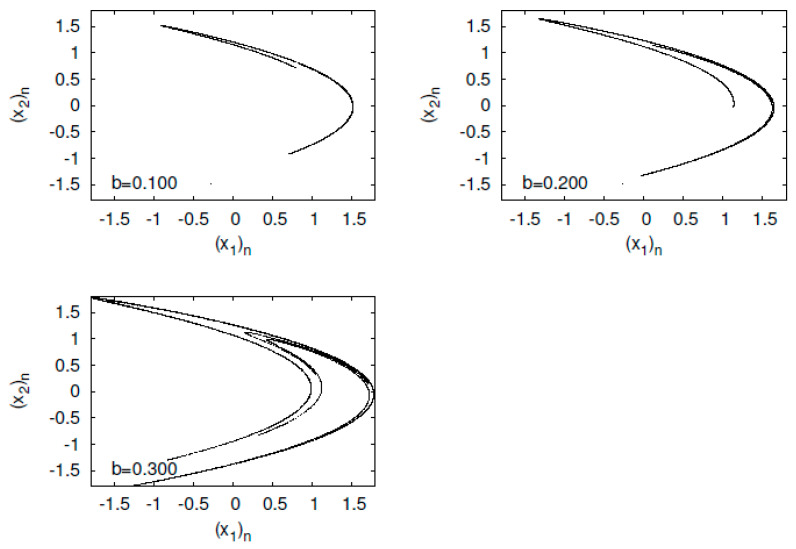
${\left(x_{2}\right)}_{n}$ versus ${\left(x_{1}\right)}_{n}$ for the Hėnon map $f_{b}$.

**Figure 8 entropy-23-01511-f008:**
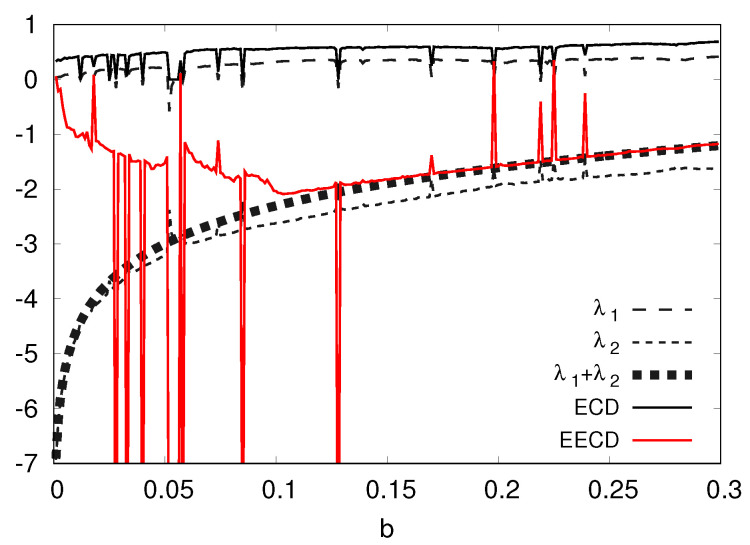
$\lambda_{1},\lambda_{2},\lambda_{1}+\lambda_{2}$, *D*, ${\tilde{D}}_{S}$ versus *a* for the Hėnon map $f_{b}$.

**Figure 9 entropy-23-01511-f009:**
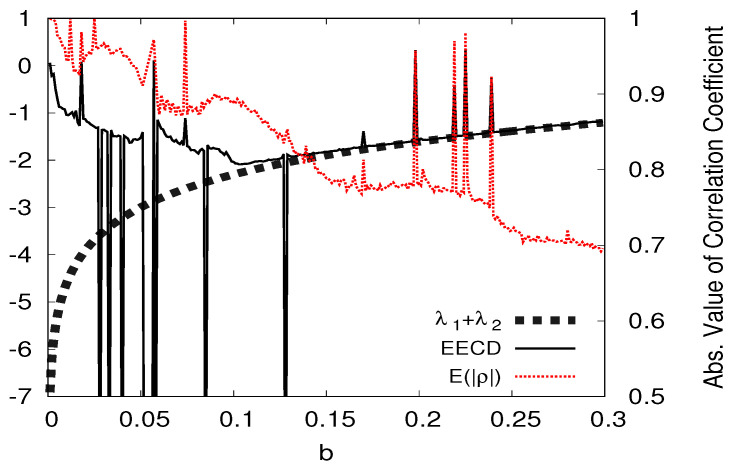
$\lambda_{1}+\lambda_{2}$, $\tilde{D}$, $E\left(\left|\rho \right|\right)$ versus *a* for the Hėnon map $f_{a}$.

**Figure 10 entropy-23-01511-f010:**
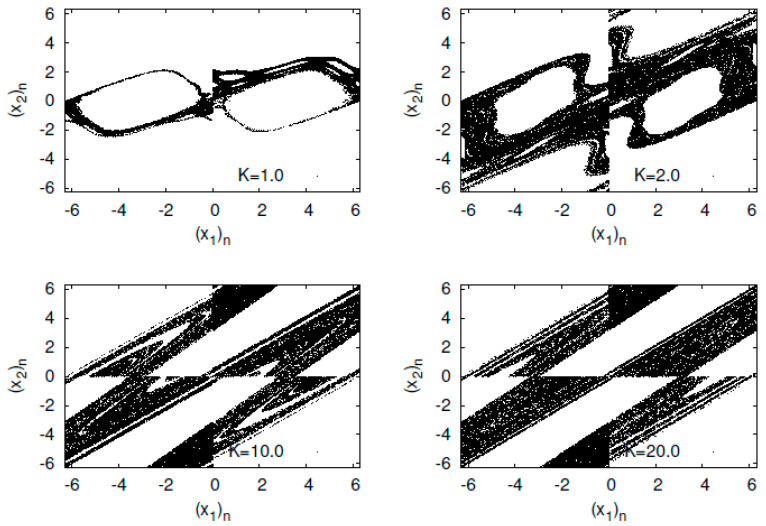
${\left(x_{2}\right)}_{n}$ versus ${\left(x_{1}\right)}_{n}$ for the standard map $f_{K}$.

**Figure 11 entropy-23-01511-f011:**
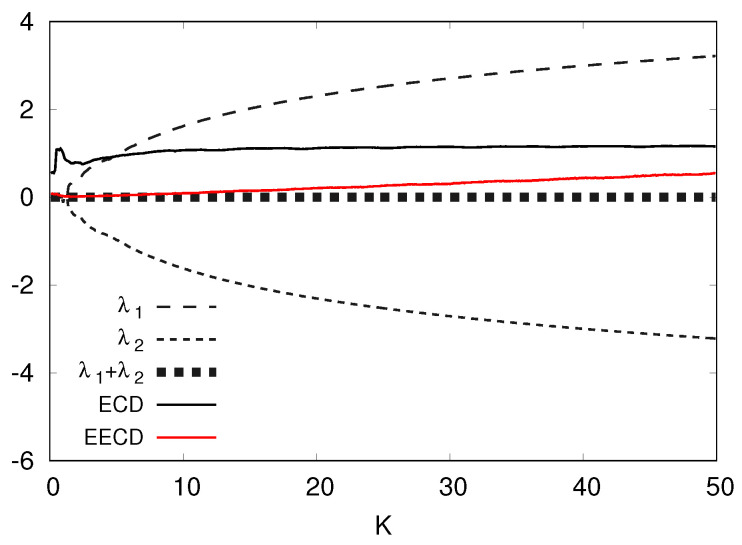
$\lambda_{1},\lambda_{2},\lambda_{1}+\lambda_{2}$, *D*, ${\tilde{D}}_{S}$ versus *K* for the standard map $f_{K}$.

**Figure 12 entropy-23-01511-f012:**
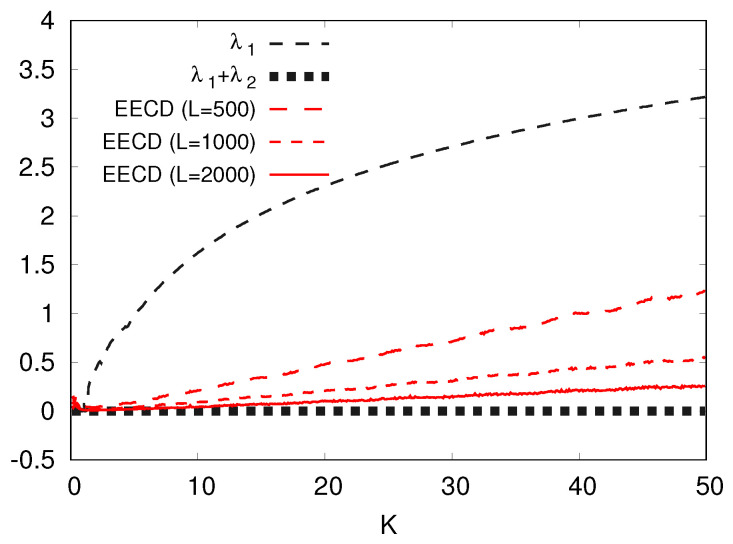
$\lambda_{1}$, $\lambda_{1}+\lambda_{2}$, ${\tilde{D}}_{S}$ at several *L*s versus *K* for the standard map $f_{K}$.

## Data Availability

Not applicable.

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
