# Peer review of "An Improved Calculation Formula of the Extended Entropic Chaos Degree and Its Application to Two-Dimensional Chaotic Maps"

_entropy, 2021, doi:10.3390/e23111511_

Round 1

Reviewer 1 Report

Comments to the revised manuscript: Manuscript ID: entropy-1435082 

All my concerns have been addressed in a satisfactory way.

However, some assumptions about the function f need to be given   (piecewise f1 ). 

The paper contains interesting results.

The manuscript has been corrected and written more precisely, so I recommend the revised manuscript for publication.

Author Response

I would like to thank the managing editor and the reviewers for their precious time and invaluable comments. I have carefully addressed all the comments.
The corresponding changes and refinements made in the revised paper are summarized in my responses.

Reviewer 2 Report

The paper under review proposes an improved calculation formula of the extended entropic chaos degree to obtain appropriate numerical computation results for two-dimensional chaotic maps.

The topic is very interesting, the results are nice but the paper is not suitable for publication in its present form. The authors should revise the paper according to the following concerns:

Main concern:

1) The authors should clarify in which cases the EECD is almost equal to the total sum of the LEs for their chaotic maps (in which cases? under which conditions? Hypotheses?). It seems that you describe the examples that work.

2) What is a stable periodic map? Please use the classic definitions.

3) The results that the author obtained for the Ikeda, Standard map (Chirikov map?) and Hénon are not well motivated. For example for the Standard map, there is a reason why the sum of the LE is zero.  This should be discussed properly.

4) The author has described the phenomena but has not explained why this happens. This should be done more carefully.

Other issues:

(i) Abstract: "In my previous work," --> which one? Please say it explicitly.

(ii) In other words, could you please define the ECD? (formula 1)

(iii)Theorem 1: "If a map f is a stable periodic" --> I am not following the result. Please rewrite.  The same at line 144.

(iv) line 71: "over the entirety of Ai."--> bad wording.

(v) Line 78: "At the end of this section, we discuss the relationship between the extended entropic chaos degree and the metric entropy." --> sometimes you write ECD, other you write explicitly. 

(vi) Formula (9) should be computed carefully. I did not follow the approximations. 

(vii) Line 106: A motivation for the generalized Baker’s map is needed. This part is missing. The same for the Tinkerbell map.

(viii) Line 133: For some parameters, the Ikeda map is conjugated to a Hénon map, right? Perhaps this should be refereed to relate sections 5.3 and 5.4.

(ix) Does the standard map has some reversing symmetry? I guess that the authors should explain why the sum of the eigenvalues for this map has the form of Figure 11. (see main concerns above)

(x) Line 208: what is L^2? 

If you take these comments seriously, I will be happy to recommend this article for publication in Entropy. 

Author Response

(The authors gave the same response as above.)

Round 2

Reviewer 2 Report

This is the second revision of the paper entitled: 

"An Improved Calculation Formula of the Extended Entropic Chaos Degree and Its Application to Two-Dimensional Chaotic Maps"

by K. Inoue.

The author has taken into account my comments and I am happy with the revision. This is not a formidable research but raise some interesting problems for future research.

This paper deserves publication in Entropy. Before publication, I would like to suggest a revision to the language since there are unclear sentences, and some repetitions. In addition, the authors should do the following minor changes:

(i) Lines 85-86 -- bad wording

(ii) Line 100: Define correctly "almost uniformly distributed over" a set. 

(iii) The information of Line 110 would be better after Line 119.

(iv) What do you mean with the the sentence on Line 131?

 (v) [just a remark for the author] Plots for the Ikeda map are very similar to the ones presented by Oksagoglu and Young:  Rank One Chaos: Theory and Applications Int. J. Bifurc. Chaos 18(5) (2008), 1261-1319. This may be stated as an open problem/ issue to be discussed later...

(vi) You have not discussed the reversing symmetry of the stardard map. At least you should refer something (for future work). The plot of Figure 11 may be explained using symmetry (reversing symmetry). I guess that the author ignored my comment on the Chirikov map. 

(vii) Please revise the English language of the manuscript. 

I would like to check the final version of the manuscript before being accepted.

Author Response

I am extremely grateful to the reviewers for their helpful comments and suggestions. 
I have carefully addressed all the comments.
The corresponding changes in the revised paper are summarized in my responses below.

Round 3

Reviewer 2 Report

I am happy with the revision. The author has considered my comments and the paper may be published. 

Some minor points:

(i) Uniformize "I" and "we" throughout the text (Line 89: I; Line 93: we);

(ii) Line 231: Since the map is reversible, then the two Lyapunov exponents (of the first return map to a cross section) satisfy the conditions of "Generic area-preserving reversible diffeomorphisms" By Bessa et al. (2015)

(iii) Line 63: "then we obtain the following."--> ", the following equality holds"

(iv) Line 153: "we obtain the following." --> "we obtain the following:"

(v) Line 165: "However, the Ikeda map f_acreates a stable periodic orbit at several a’s. If the Ikeda map fa creates a stable periodic orbit," --> repetition.

(vi) Line 221: Add the reference by Bessa et al (2015): "Generic area-preserving reversible diffeomorphisms". 

If the authors correct my points, I recommend the paper for publication. There is no need to check the paper once again.

Author Response

I am extremely grateful to the reviewers for their helpful comments and suggestions. 
I have carefully addressed all the comments.
The corresponding changes in the revised paper are summarized in my responses below.

This manuscript is a resubmission of an earlier submission. The following is a list of the peer review reports and author responses from that submission.

Round 1

Reviewer 1 Report

The manuscript treats about a nonlinear dynamical tool to characterize the chaotic features for a given time-series, namely Extended Entropic Chaos Degree.

The manuscript is well written and the results are well described. I believe that the manuscript deserves publication.

Reviewer 2 Report

July 7, 2021

Review of the paper:

Improvement of the Computation Method of the Extended Entropic Chaos Degree for Quantifying Chaos in a Conservative System

by Kei Inoue

Submitted to Entropy

General Comments

In this paper the Author develops the concept of entropic chaos degree (ECD) and introduced the concept of extended entropic chaos degree. He argues that the computational performance of his concept is better than ECD.

Major Comments

I find the idea presented in the paper interesting, but I have the following comments and concerns: 

  1. In the Abstract it should be included, even in a brief form, what is advantage of the (computational) method proposed, what is the added value of the paper.
  2. The Author wrote that Extended Entropic Chaos Degree can be used for quantifying chaos. In my opinion a definition of chaos should be recalled which is considered by the Author in this paper.
  3. It would be more convincingly to present the graphs of Lyapunov Exponent (LE) and Extended Entropic Chaos Degree (EECD) in the same Figure. Some more advanced comparison of EECD and LE should be included and not only optical one comparison.
  4. What is new in this paper when compare to the paper announced in arXiv:

APPLICATION OF CHAOS DEGREE TO SOME DYNAMICAL SYSTEMS

by Kei Inoue, Masanori Ohya and Keiko Sa.

  1. Some paragraph should be included describing the relations between ECD and well known in the literature, since many years, the concepts of „topological entropy” and „metric entropy”.
  2. The connection between Lyapunov Exponent and ECD should be described in a more detailed and precise way.
  3. Relation to the „Discrete Lyapunov Exponent” as proposed in the paper by Kocarev and Szczepanski published in Physical Review Letters (2004) should be commented/addressed.
  4. Some assumption concerning the function f should be given. Is this a "smooth" function or it is "measurable" function only, or ...? Do this f has invariant measure ?

Minor Comments

A) More adequate abbreviations should be used to the concepts that appear in the paper, e.g. entropic chaos degree would be associated with the abbreviation ECD rather than entropic (CD) or so. Similarly to the concept extended entropic chaos degree maybe the abbreviation EECD is more adequate.

B) In my opinion English should be improved.

Final Comments

The article contains interesting considerations and seems to be of some value. However, I found some missed points which are listed above. Therefore, I recommend publication of this paper provided that the above concerns and problems will be carefully addressed. At this moment I recommend Major Revision.

Reviewer 3 Report

The Lyapunov exponents (LE) are used to quantify the chaos in the dynamical systems theory.  Nevertheless, it is (in general)  difficult to compute the LE of dynamical systems from a time series. The authors proposed an improvement of the entropic chaos degree as a criterion for quantifying chaos in dynamical systems through information dynamics.

The author claims that the method is directly computable for any time series associated to a conservative system (see my comment (vi)). The topic is interesting and I think that the paper gives a contribution to the field. However, the manuscript is not suitable for publication as it stands. I may recommend the publication of the article if the author are prepared to incorporate the following changes/improvements:

(i) Lines 32-34: the sentence with definitions and terminology is cryptical. Please rewrite. 

(ii) Line 42: what is a repartition? Author should define correctly. These definitions could be found in any reference of Ergodic Theory (when developing Entropy).

(iii) Assumption 1(2) is really necessary? Could you please explain it better (in other words).

(iv) Please relate definition of Line 48 with formula (6).

(v) In Section 4.1, a dissipative system is analysed. In Section 4.2 a conservative system is studied. A good motivation for the study of these two cases is compulsory.

(vi) In section 5, it seems that the method is valid for conservative and dissipative systems. Is it right? What are the main differences? In lines 172-174, once again, the method is applicable in both settings. Am I wrong?

(vii) In (13), please correct the formula. "A_i :" instead of "A_i|". The same in (14).

(viii) Line 102: please explain the inequalities with more details. 

(ix) Line 104: please motivate why are you taking M=1,000,000.

(x) Figure 5: within the "tips" of the graph, what are you expecting to obtain? What is the meaning of the r_{input} (notice that I am not asking for the definition). The sentence of Line 161 is not clear to me. 

(xi) Line 140: I get lost in "Therefore....". Why? This should be described with  more detail.

(xii) Line 176: typo "Workds" 

(Xiii) A description of the dynamical structures of the Hénon map (as well as the Chirikov map) is missing. The relation of these new definitions and Metric entropy should be discussed. 

(xiv) I think that the question raided in the abstract " (...) it is difficult to compute the Lyapunov exponent of dynamical systems from a time series" has not been replied.  What is the role of this manuscript to answer the question? This should be carefully discussed.

At first glance, your paper seems to be outstanding. But when we read the details, it is almost a slight generalisation.  Please go deeper with the details and explain carefully the settings/proofs. The answers to  comments (v)-(vi)- (xiv) will be particularly important.